Comparative efficacy and safety of 180 W XPS vs. 120 W HPS GreenLight laser therapy for benign prostatic hyperplasia: a systematic review and meta-analysis

Wang Min 819203381@qq.com 1
Xi Yu 2
Qiu Zan 3
Huang Nanxiang 1
Zhang Li 1
Liu Jinlong 1
1 Department of Paediatric Surgery, Nanchong Central Hospital , Nanchong , Sichuan , China
2 Department of Operating Room, Nanchong Hospital of Traditional Chinese Medicine , Nanchong , Sichuan , China
3 Department of Sichuan Medical Science and Technology Education Center, Sichuan Medical Science and Technology Education Center , Chengdu , Sichuan , China
Marunaka Yoshinori
Electronic publication date: 2024 Nov 27
Publication date: 2024
Volume: 12
Electronic Location ID: e18615
Received 2024 Jun 17; Accepted 2024 Nov 8
Copyright: ©2024 Wang et al.
Copyright year: 2024
Copyright holder: Wang et al.
License: This is an open access article distributed under the terms of the Creative Commons Attribution License, which permits unrestricted use, distribution, reproduction and adaptation in any medium and for any purpose provided that it is properly attributed. For attribution, the original author(s), title, publication source (PeerJ) and either DOI or URL of the article must be cited.
License URL: https://creativecommons.org/licenses/by/4.0/

Keywords: Prostatic hyperplasia, Greenlight, XPS, Vaporization, Surgical outcomes

Funding: The authors received no funding for this work.

==============================
Objectives

To compare the surgical and functional outcomes of the 120 W HPS and 180 W XPS GreenLight laser vaporization systems in the treatment of benign prostatic hyperplasia (BPH).

Methods

In January 2024, a comprehensive search across PubMed, Cochrane Library, and EMBASE was conducted following PRISMA guidelines, registered under PROSPERO (CRD42024531176). Studies comparing 120 W and 180 W GreenLight lasers in BPH treatment were assessed for clinical outcomes.

Results

Eight studies were included. The 180 W XPS system improved operation time (MD: 12.70, 95% CI [5.29–20.11], p = 0.0008), lasing duration (MD: 10.09, 95% CI [0.85–19.33], p = 0.03), and catheterization duration (MD: 0.43, 95% CI [0.12–0.74], p = 0.007). No significant differences in energy consumption, energy density, or length of hospital stay were found. Functional outcomes such as International Prostate Symptom Score and maximum urinary flow rate showed no significant differences, except in quality of life (MD: 0.43, 95% CI [0.06–0.80], p = 0.02) and prostate-specific antigen levels (MD: −0.77, 95% CI [−1.28 to −0.25], p = 0.003). The 180 W system exhibited a lower rate of overall (OR: 1.52, 95% CI [1.14–2.04], p = 0.005) and minor complications (OR: 1.84, 95% CI [1.27–2.66], p = 0.001), with no significant differences in major complications or other adverse events.

Conclusions

The 180 W XPS system demonstrates enhanced efficiency and reduced complication rates, offering a favorable option for BPH treatment, particularly for larger prostates. Future studies should focus on randomized trials to confirm these findings and assess long-term outcomes.

Introduction

Benign prostatic hyperplasia (BPH) is a prevalent urological condition in middle-aged and elderly men, with incidence rates increasing with age (Calogero et al., 2019). BPH primarily manifests as lower urinary tract symptoms (LUTS) due to bladder outlet obstruction and increased smooth muscle tone (Calogero et al., 2019; Rosen, Giuliano & Carson, 2005). While medications provide early relief, surgical intervention is required for those unresponsive to pharmacological treatments (Gravas et al., 2020; Parsons et al., 2020).

Historically, transurethral resection of the prostate (TURP) has been the gold standard for minimally invasive BPH surgery since its introduction in 1932 (Gravas et al., 2020; Parsons et al., 2020). However, advancements in medical technology have led to various minimally invasive techniques, notably laser therapies. One of the primary modalities of this therapy is the GreenLight laser, which operates by emitting a 532 nm wavelength laser highly absorbed by hemoglobin, allowing for precise tissue vaporization while simultaneously coagulating blood vessels to minimize bleeding (Gomez-Sancha, 2015). Although holmium and thulium laser enucleation procedures have gained popularity in recent years, for patients undergoing antiplatelet or anticoagulant therapy, the GreenLight laser remains the most commonly used technique, accounting for 39% of procedures, which is significantly higher than holmium and thulium lasers, which are used in only 12% of cases (Becker et al., 2017; Chen et al., 2022). GreenLight laser vaporization has emerged as a viable TURP alternative, reducing postoperative catheterization, hospitalization durations, and minimizing complications, especially in patients with coagulation disorders (Gill et al., 2024; Ma et al., 2022). Initially launched with the 80 W KTP laser, which had limitations in tissue ablation speed and extent, particularly for larger prostates, enhancements were made with the 120 W HPS system (Al-Ansari et al., 2010; Hueber et al., 2012). The 120 W system improved focus and power density, reducing operational times and increasing ablation efficiency (Kang et al., 2008; Te, 2006). This system offered improved focus and power density, reducing operational times and increasing ablation efficiency (López et al., 2016). The evolution of GreenLight laser systems has significantly improved the efficacy of photoselective vaporization of the prostate (PVP).

Despite the introduction of the 180 W XPS system, which offers increased power density and fiber design improvements for enhanced tissue vaporization efficiency and reduced operative times, systematic evidence comparing the 180 W and 120 W systems remains limited (Brunken, Seitz & Woo, 2015). Existing studies suggest that the 180 W system shows similar short-term safety and efficacy to the 120 W system in early clinical trials, but comprehensive comparisons of long-term outcomes and patient recovery are lacking (Lichy et al., 2022; Meskawi et al., 2017). Therefore, we conducted a meta-analysis to systematically evaluate and compare the clinical efficacy and safety of GreenLight 120 W HPS versus 180 W XPS vaporization in BPH treatment.

Materials & Methods

This systematic review and meta-analysis strictly adheres to the Preferred Reporting Items for Systematic Reviews and Meta-Analyses (PRISMA) guidelines. The study was prospectively registered with the PROSPERO database (University of York, York, UK) under the unique identifier CRD42024531176. It is important to note that the preliminary screening of search results, conducted to refine the study’s scope and confirm its feasibility, occurred before our formal registration. This preliminary screening was part of the preparatory work and did not involve data analysis or collection, ensuring our study’s compliance with requirements for prospective registration.

Literature search

In January 2024, a comprehensive literature review was executed using databases such as PubMed, Cochrane Library, and EMBASE, employing search terms including ‘prostate’, ‘120 W’, ‘180 W’, and ‘laser vaporization’. The search was conducted without limitations on language or publication timeline. Additionally, a meticulous manual search of references from relevant articles was performed via PubMed to ensure an exhaustive collection of pertinent data.

The inclusion criteria were studies that focused on patients diagnosed with BPH and reported comparative outcomes between the 120 W and 180 W GreenLight laser treatments. The exclusion criteria were studies that dealt with conditions other than BPH, those that did not provide direct comparisons between the specified laser powers, and documents categorized as letters, case reports, commentaries, conference abstracts, or studies with incomplete or irrelevant data concerning the effectiveness and safety of the treatments.

The search strategy was meticulously executed by two authors (M.W and Y.X). Both authors independently conducted the literature search and initially screened the results based on predefined eligibility criteria. Disagreements between the two reviewers were resolved through discussion. If a consensus could not be reached, a third reviewer (NX.H), served as the referee to make the final decision. This process ensured a thorough and unbiased selection of studies for inclusion in the analysis.

Quality assessment

Quality evaluation of the included studies was conducted using the Newcastle-Ottawa Scale (NOS), with a score of 7 or higher indicating high quality. Additionally, the risk of bias in each study was independently analyzed using the Risk of Bias in Non-Randomized Studies of Interventions (ROBINS-I) framework, which is designed specifically for the appraisal of non-randomized studies of interventions.

Data extraction

Data extraction encompassed a detailed review of crucial elements from each included study, including the authors, publication year, study design, sample size, and specific preoperative data such as age, prostate volume, International Prostate Symptom Score (IPSS), quality of life (QoL) scores, maximum urinary flow rate (Qmax), postvoid residual urine volume (PVR), and prostate-specific antigen (PSA) levels. Outcome metrics analyzed included operation time, energy consumption, energy density (calculated as the energy expended per volume of prostate tissue removed), lasing duration, number of laser fibers used, length of hospital stay (LOS), duration of postoperative catheterization, and improvements in IPSS, QoL, Qmax, PVR, and PSA levels after surgery. Comprehensive data on postoperative complications were also compiled, including overall complications and specific adverse events such as capsular perforation, urinary retention, dysuria, re-treatment rates, bladder neck contracture, and urethral stenosis. Furthermore, data on minor and major complications, classified according to the Clavien-Dindo grading, were meticulously gathered. Specifically, ‘minor’ complications referred to Grade I and II events, while ‘major’ complications encompassed those classified as Grade III–V (Clavien et al., 2009; Dindo, Demartines & Clavien, 2004).

Statistical analysis

Statistical evaluations were conducted using Review Manager (RevMan) software, version 5.3. Continuous outcomes were analyzed for mean differences (MD), and dichotomous outcomes were assessed through odds ratios (OR), both reported with 95% confidence intervals (CIs). Heterogeneity among studies was quantified using chi-squared and I-squared tests. In cases where significant heterogeneity was detected (I2 > 50%), analyses were performed using random-effects models, whereas fixed-effects models were employed for low heterogeneity scenarios. Statistical significance was determined by a p-value less than 0.05. Additionally, sensitivity analyses were executed using Stata software, version 17, to explore the robustness of the findings and investigate potential sources of heterogeneity.

Results

After an exhaustive screening process involving the exclusion of duplicate entries, elimination of irrelevant records, and rejection of studies due to substandard methodologies or inadequate data disclosure, a total of eight studies (Ben-Zvi et al., 2013; Campbell et al., 2013; Eken et al., 2015; Hueber et al., 2013; Inaba et al., 2023; López et al., 2016; Rieken et al., 2013; Tanchoux et al., 2012) were ultimately selected for our systematic review. This thorough filtering process is illustrated in Fig. 1. Table 1 provides a systematic compilation of the baseline characteristics and quality evaluations of these studies. Risk of bias was assessed using the ROBINS-I tool, revealing one study as high risk due primarily to incomplete data reporting and the presence of confounding variables. The rest were deemed to have a moderate risk of bias, as shown in Fig. 2. Considering the significant impact of prostate volume on surgical and functional outcomes, we first investigated whether there was a difference in preoperative prostate volume between the HPS and XPS groups. The results showed no statistically significant difference in preoperative prostate volume between the two groups, as shown in Fig. S1.

Figure 1 PRISMA flow diagram.

Surgical outcomes

The 180 W XPS system was found to have advantages in operation time (MD: 12.70, 95% CI [5.29–20.11] p = 0.0008, Fig. 3A), lasing duration (MD: 10.09, 95% CI [0.85–19.33], p = 0.03, Fig. 3B) and duration of postoperative catheterization (MD: 0.43, 95% CI [0.12–0.74], p = 0.007, Fig. 3C) when compared to 120 HPS system. However, no significant differences observed in energy consumption (MD: −55.10, 95% CI [−127.78–17.57], p = 0.14, Fig. 4A), energy density (MD: −0.60, 95% CI [−2.24–1.05], p = 0.48, Fig. 4B), number of laser fibers used (MD: 0.56, 95% CI [−0.18–1.30], p = 0.14, Fig. 4C), and LOS (MD: 0.57, 95% CI [−0.33–1.46], p = 0.21, Fig. 4D).

Functional outcomes

No significant differences were observed in IPSS (MD: 1.31, 95% CI [−0.27–2.89], p = 0.10, Fig. 5A), Qmax (MD: 0.54, 95% CI [−3.93–5.02], p = 0.81, Fig. 5B) and PVR (MD: 5.55, 95% CI [−67.41–78.51], p = 0.88, Fig. 5C), except for QoL (MD: 0.43, 95% CI [0.06–0.80], p = 0.02, Fig. 5D) and PSA (MD: −0.77, 95% CI [−1.28 to −0.25], p = 0.003, Fig. 5E) between the two approaches. The improvement in QoL scores and PSA reduction in the 180 W XPS system reflects a better overall patient satisfaction and potential for reducing prostate volume, which may further translate to long-term benefits in BPH management.

Complications

A lower rate of overall complications (OR: 1.52, 95% CI [1.14–2.04] p = 0.005), minor complications (OR: 1.84, 95% CI [1.27–2.66], p = 0.001), and urinary retention (OR: 1.80, 95% CI [1.02–3.16], p = 0.04) were associated with the 180 W XPS. No significant differences observed in major complications (OR: 0.54, 95% CI [0.04–7.16], p = 0.64), capsular perforation (OR: 0.52, 95% CI [0.20–1.37], p = 0.18), dysuria (OR: 1.16, 95% CI [0.67–2.00], p = 0.59), re-treatment rates (OR: 2.36, 95% CI [0.68–8.19], p = 0.17), bladder neck contracture (OR: 0.64, 95% CI [0.12–3.35], p = 0.60), and urethral stenosis (OR: 0.91, 95% CI [0.12–6.68], p = 0.93). These observations on specific postoperative complications are detailed in Table 2.

Table 1 Characteristics and quality assessment of included studies.

Author, year	Study design	No. of patients	Age (years)	Prostate volume (mL)	IPSS	QoL	Qmax (mL/s)	PVR (mL)	PSA (ng/dL)	Quality scoree	
Ben-Zvi et al. (2013)	P	80/120	69.2 (48–87)/67.9 (50–85)a	80.3 (33–187)/79.1 (31–229)	25.4 (15–35)/24.2 (16–34)	4.8 (3–6)/4.5 (3–6)	7.2 (0–17)/7 (2–13)	280 (54–620)/308 (51–1350)	4.8 (0.3–14)/4.2 (0.3–19)	8	
Campbell et al. (2013)	P	50/50	68 (60–73.8)/66.5 (60–71.8)b	51 (37.8–72.3)/68 (45.5–94)	21 (17–26)/20 (14–25)	4 (4–5)/4 (4–5)	9 (7–11)/9 (6.2–12.2)	110 (74.5–195.5)/143 (63–260)	3.4 (1.7–4.9)/3.55 (1.7–7.3)	7	
Eken et al. (2015)	P	88/73	70.2 (50–83)/68.55 (49–86)a	62.32 (28–128)/61.3 (27–142)	24.83 (16–32)/23.73 (15–33)	4.55 (3–6)/4.58 (3–6)	7.24 (4.8–11.4)/7.38 (5.3–12.4)	320 (35–740)/305 (30–700)	2.93 (0.5–10.2)/2.79 (0.4–15.1)	8	
Hueber et al. (2013)	P	1187/622	71.54 ± 10/68.93 ± 10c	76.66 ± 48/70.1 ± 50	NA	NA	NA	NA	NA	7	
Inaba et al. (2023)	R	86/86	72 (58–90)/72 (53–89)d	120 (100–250)/124 (100–209)	21.5 (5–35)/22 (4–34)	5 (2–6)/6 (3–6)	6.3 (2.3–21.3)/7.6 (2.3–28.4)	NA	8.2 (0.8–157)/8.1 (0.8–46.1)	7	
López et al. (2016)	R	109/82	70.5 ± 7.9/69.9 ± 12.3	54.2 ± 18.3/53.9 ± 20.9	22.6 ± 5.1/24.6 ± 5.9	4.86 ± 0.9/4.89 ± 1.1	8.1 ± 3.2/13.1 ± 2.2	NA	3.7 ± 3.3/4.3 ± 5.9	7	
Rieken et al. (2013)	R	80/80	72 ± 11/73 ± 8	64 ± 46/64 ± 36	21 ± 6/15 ± 6	4 ± 2/3 ± 1	14.1 ± 14.3/8.9 ± 4.5	161 ± 125/88 ± 141	6 ± 8.3/5.3 ± 8.8	8	
Tanchoux et al. (2012)	P	25/25	70.9 ± 7.8/73.6 ± 8.2	77 ± 24/72 ± 18	17 ± 8/16 ± 5	NA	8.7 ± 4.1/7.3 ± 3.2	NA	4.5 ± 3.1/5.4 ± 3.7	8	
Notes.

/ 120 W HPS versus 180 W XPS

P Prospective

R Retrospective

IPSS International Prostate Symptom Score

QoL quality of life score

Qmax maximum urinary flow rate

PVR postvoid residual urine volume

PSA prostate-specific antigen

NA not available

a Mean (range).

b Median (Interquartile range, IQR).

c Mean ± Standard Deviation, SD.

d Median (range).

e Using NOS scoring Rules.

Figure 2 The risk of bias for the included studies according to ROBINS-I (Ben-Zvi et al., 2013; Campbell et al., 2013; Eken et al., 2015; Hueber et al., 2013; Inaba et al., 2023; López et al., 2016; Rieken et al., 2013; Tanchoux et al., 2012).

Figure 3 Forest plot and meta-analysis of operation time (A), lasing duration (B), duration of postoperative catheterization (C).

Figure 4 Forest plot and meta-analysis of energy consumption (A), energy density (B), number of laser fibers used (C), and LOS (D).

Figure 5 Forest plot and meta-analysis of improvements in IPSS (A), Qmax (B), PVR (C), QoL (D) and LOS (E).

Sensitivity analyses

The analysis confirmed the stability of our findings, indicating that the observed heterogeneity did not significantly affect the overall conclusions of the meta-analysis. Further sensitivity analysis showed consistent results across different study subgroups, reinforcing the robustness of the findings. Details of these analyses are provided in the Supplementary Materials.

Discussion

Although the PVP procedure using the 120 W laser can be conducted safely, studies report retreatment rates as high as 9–16% in patients with larger prostates necessitating significantly longer laser and operation times, highlighting the need for improved energy delivery (Al-Ansari et al., 2010; Hueber et al., 2012; Tasci et al., 2011). To enhance the efficiency, efficacy, and durability of PVP treatments, the 180 W XPS system has been introduced, specifically for prostate volumes over 80 mL. This system, with increased power and improved fiber optics, aims to enhance tissue ablation efficiency and achieve more uniform and rapid energy distribution, potentially reducing operative times and improving patient outcomes (Ben-Zvi et al., 2013). Additionally, Marchioni et al. (2018) reported that the incidence of major acute cardiovascular events following 180 W GreenLight PVP was only 1.9% in a multicenter retrospective study. Furthermore, evidence suggests that GL-180-W XPS PVP can be safely performed alongside concomitant endoscopic or open/laparoscopic surgeries (Castellucci et al., 2020). However, in the absence of direct comparative studies between the 120 W HPS and 180 W XPS systems, our meta-analysis aims to address this gap by systematically evaluating their clinical outcomes.

Table 2 Results of meta-analysis of postoperative complications.

Outcomes	No. of studies	No. of patients	Heterogeneity	Merge analysis	
			Chi2	I2(%)	Pvalue	OR (95% CI)	P value	
Overall complications	7	518/516	6.56	9	0.36	1.52 (1.14–2.04)	0.005	
Minor complications	5	344/357	4.72	15	0.32	1.84 (1.27–2.66)	0.001	
Major complications	2	130/130	2.84	65	0.09	0.54 (0.04–7.16)	0.64	
Specific adverse events								
Urinary retention	6	493/491	2.44	0	0.79	1.80 (1.02–3.16)	0.04	
Capsular perforation	3	210/250	0.59	0	0.75	0.52 (0.20–1.37)	0.18	
Dysuria	3	248/273	0.20	0	0.90	1.16 (0.67–2.00)	0.59	
Re-treatment	4	363/361	1.14	0	0.77	2.36 (0.68–8.19)	0.17	
Bladder neck contracture	2	138/123	1.20	17	0.27	0.64 (0.12–3.35)	0.60	
Urethral stenosis	2	138/123	0.77	0	0.38	0.91 (0.12–6.68)	0.93	
Notes.

No. number

/ 120 W HPS versus 180 W XPS

Chi2 Chi-square test value

I2 I-squared test value

OR odds ratios

95% CI 95% confidence interval

In this meta-analysis, we identified significant advantages of the 180 W XPS system over the 120 W HPS system in operation time, lasing duration, and postoperative catheterization. These findings highlight the superior efficiency of the 180 W XPS system, especially beneficial for larger prostate volumes where the 120 W HPS system has limitations. Equipped with the MoXy fiber, which features a larger core and advanced cooling capabilities, the 180 W XPS offers improved energy delivery (Zorn & Liberman, 2011). This enhancement likely accounts for the reduced operation and lasing times, indicating a more efficient tissue vaporization process (Ben-Zvi et al., 2013; Inaba et al., 2023; Rieken et al., 2013). Despite its increased power setting, the 180 W XPS system achieves these efficiencies without proportional increases in overall energy consumption, balancing power output and operational duration. Scatter plots by Ben-Zvi et al. (2013) showed that the 180 W XPS, equipped with MoXy fiber technology, provides a faster and more predictable vaporization rate, especially for prostates larger than 75 mL. The influence of surgical technique on tissue ablation efficiency was evident. Ex vivo bovine model experiments indicated that vaporization efficiency is compromised with faster and wider sweeping angles, suggesting that skilled surgeons can achieve greater tissue removal with the same energy in less time (Kauffman, Kang & Choi, 2009; Osterberg et al., 2011). Optimal fiber-to-tissue distance ensures high power density due to the shallow penetration of the 532 nm laser beam, focusing energy on vaporization. Excessive distance, particularly with lower power lasers like the 80 W, decreases power density and increases coagulation risks (Rieken et al., 2013). Preclinical data from in vitro studies on porcine kidney tissues corroborate these findings, showing a 62%–75% faster vaporization rate with the 180 W XPS compared to the 120 W HPS. This efficiency expedites the procedure and creates significantly larger cavities, improving surgical outcomes and reducing operative times (Liatsikos et al., 2012; Malek et al., 2011).

Furthermore, our analysis showed no significant differences in energy consumption or energy density between the two systems. This suggests that the enhanced power of the 180 W XPS translates to more efficient energy application rather than increased energy use per procedure. This efficiency likely contributes to the similar number of laser fibers used and LOS reported for both systems, indicating that while the 180 W XPS system improves specific procedural aspects, it does not alter broader resource utilization patterns. These insights into the performance differences underscore the importance of continuous technological advancements in laser systems. However, the findings caution against assuming direct improvements in all procedural aspects due to technological upgrades. Variability in surgical outcomes, influenced by surgeon experience and patient characteristics, means these results should be considered within a broader clinical context. Future studies should aim for randomized controlled trials and prospective studies to validate these findings and potentially standardize operative approaches, enhancing the generalizability and clinical applicability of the newer 180 W XPS system in treating BPH.

In our meta-analysis, improvements in functional outcomes following prostate vaporization procedures were assessed, revealing no significant differences between the two surgical approaches for IPSS, Qmax, and PVR, except for QoL scores and PSA levels. Specifically, QoL and PSA showed statistically significant improvements favoring the enhanced laser system. These results suggest that while overall urinary function and symptomatology may not differ markedly between systems, the quality of life improvements and biochemical markers indicate more effective tissue removal and potentially better clinical outcomes with the newer system. The nuanced differences in functional outcomes, particularly the QoL improvements, could reflect the more efficient and thorough tissue vaporization provided by the updated laser technology (Inaba et al., 2023). A reduction in PSA levels exceeding 50% indicates effective prostate tissue removal (Valdivieso et al., 2016). The more pronounced decrease in PSA levels supports this, indirectly affirming the efficacy of the 180 W XPS system. Despite these advantages, the substantial similarities in IPSS, Qmax, and PVR outcomes across both systems highlight their effectiveness in managing the primary functional symptoms of BPH. Interestingly, Rieken and colleagues reported more pronounced improvements in IPSS, QoL, PVR volume, and Qmax in the 120 W group compared to the 180 W group postoperatively, although their study noted significantly higher baseline levels and a shorter follow-up period in the 120 W group (Rieken et al., 2013). This necessitates longer functional follow-ups to draw definitive conclusions. Future research should continue to explore these outcomes over extended periods to better understand the long-term benefits and potential differences in reoperation rates between these technologies.

The transition to the 180 W XPS laser involves a switch to a larger diameter fiber (750 µm vs. 600 µm), which can reduce the endoscopic field of view due to the increased fiber size obstructing more of the visual field. However, the larger fiber also allows for greater irrigation flow, which is critical for maintaining a clear surgical field by flushing out debris and blood more efficiently (Campbell et al., 2013; Hueber et al., 2013). Campbell et al. (2013) studied 100 patients, with the first 50 treated using the 120 W HPS system and the latter 50 with the 180 W XPS system. They found that the visual impacts of the 180 W system did not affect complication rates, indicating that surgeons can effectively adapt to the XPS system, with similar learning curves for both systems (Campbell et al., 2013). In our meta-analysis, we observed a notable reduction in overall and minor complications, particularly urinary retention, with the 180 W system, supporting its improved safety profile. This can be attributed to the advanced laser characteristics of the XPS system, including higher power settings and the innovative MoXy fiber design, which provides efficient energy delivery and a consistent vaporization rate. The 180 W laser’s increased power, combined with the MoXy fiber’s active cooling cap technology and steel-tip cap, helps maintain fiber integrity and minimize devitrification, resulting in precise tissue ablation and reducing complications related to over-vaporization or inadequate tissue removal (Ben-Zvi et al., 2013; Hueber et al., 2013; Rieken et al., 2013; Tanchoux et al., 2012). Clinical experiences highlight that the increased power of the 180 W laser may require modified surgical techniques to prevent capsular perforation, a complication arising from deeper tissue vaporization and larger beam diameters (Campbell et al., 2013). However, our meta-analysis showed no significant differences in capsular perforation between the 120 W and 180 W systems, suggesting that such complications are not inherently higher with increased power. Surgeons need to maintain a precise distance of 3–5 mm between the fiber tip and prostate tissue to mitigate the risk of perforation and bleeding when operating near the capsule (Rieken et al., 2013). This underscores the importance of experience and adaptability in managing higher-powered lasers effectively. Despite these precautions, capsular perforation occurrences are sporadic and not necessarily more frequent with high-power systems, provided power density is carefully managed (Campbell et al., 2013; Rieken et al., 2013). This suggests that with adequate surgical expertise and careful application of laser settings, the risk of significant complications can be minimized, even with high-power lasers.

Regarding the cost-effectiveness of the GreenLight XPS system, literature suggests economic benefits (Benejam-Gual et al., 2014). Studies indicate that the XPS system, by reducing fiber usage and operating times, could lower overall treatment costs (Hueber et al., 2013; López et al., 2016; Rieken et al., 2013). The 180 W XPS laser, supporting up to 400 kJ per fiber, allows for more energy application per case compared to the 120 W and 80 W systems, potentially decreasing the number of fibers needed per surgery (Rieken et al., 2013). These factors, along with outpatient management and streamlined preoperative planning using transrectal ultrasonography, suggest that the XPS system might enhance procedural efficiency and reduce hospitalization costs, making it a cost-effective alternative in publicly funded healthcare settings (Ben-Zvi et al., 2013). Additionally, current research has paid limited attention to postoperative sexual function differences between the two laser treatments. Only one study reported changes in the index of erectile function (IIEF) scores between groups, finding no statistically significant differences (Campbell et al., 2013). Similarly, an animal study found no differences in the incidence of new erectile dysfunction between the groups (Malek et al., 2011). Future research is needed to explore variations in costs and sexual function outcomes extensively.

This study’s limitations stem primarily from its reliance on observational studies, both prospective and retrospective, which may introduce biases and reduce result accuracy. Despite employing random effects models to address significant heterogeneity, some results still exhibited high heterogeneity, potentially affecting the stability and generalizability of our findings. Additionally, our analysis may be subject to publication bias, as studies with unpublished or negative results might not have been included. These limitations could impact the reliability and comprehensiveness of our conclusions, suggesting that future research should utilize more rigorous study designs and methodological strategies to enhance the quality of evidence.

Conclusions

The meta-analysis presented a comparative evaluation of the GreenLight 120 W HPS and 180 W XPS vaporization systems for the treatment of BPH, revealing that the 180 W XPS system provides significant advantages in reducing operation time, lasing duration, and postoperative catheterization duration. Despite equivalent energy consumption and no observed differences in major complications such as capsular perforation, the 180 W system showed a reduction in overall and minor complications, particularly urinary retention. Notably, quality of life improvements and a more pronounced decrease in PSA levels with the 180 W system suggest a more effective tissue removal compared to the 120 W system. These enhancements make the 180 W XPS a preferable option, particularly for managing larger prostates, emphasizing the benefits of technological advancements in laser surgery. However, the clinical implications of these findings warrant cautious interpretation due to the observational nature of included studies and high heterogeneity in some outcomes. Future research should focus on randomized controlled trials and long-term follow-up studies to further validate these findings and refine laser treatment protocols for BPH, ensuring broader application and maximized patient benefits.

Supplemental Information

Supplemental Information 1 PRISMA checklist

Supplemental Information 2 Rationale and Contribution to Knowledge

Supplemental Information 3 Supplementary Figures

Supplemental Information 4 Raw data

Supplemental Information 5 Codebook

Additional Information and Declarations

Competing Interests

Author Contributions

Data Availability

The authors declare there are no competing interests.

Min Wang conceived and designed the experiments, prepared figures and/or tables, authored or reviewed drafts of the article, and approved the final draft.

Yu Xi conceived and designed the experiments, authored or reviewed drafts of the article, and approved the final draft.

Zan Qiu conceived and designed the experiments, performed the experiments, analyzed the data, authored or reviewed drafts of the article, and approved the final draft.

Nanxiang Huang performed the experiments, analyzed the data, prepared figures and/or tables, and approved the final draft.

Li Zhang performed the experiments, analyzed the data, prepared figures and/or tables, and approved the final draft.

Jinlong Liu performed the experiments, analyzed the data, prepared figures and/or tables, and approved the final draft.

The following information was supplied regarding data availability:

This is a systematic review/meta-analysis.

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
