# Peer review of "Comparative efficacy and safety of 180 W XPS vs. 120 W HPS GreenLight laser therapy for benign prostatic hyperplasia: a systematic review and meta-analysis"

_PeerJ, doi:10.7717/peerj.18615_

## Round 0.1 · original submission · Major Revisions

Please revise your manuscript according to the reviewers' comments.
Yours,
Yoshi
Prof. Yoshinori Marunaka, M.D., Ph.D.

·

Basic reporting

The authors should be complimented for their work. The scientific soundness of this study is good and the route it traces to future clinical decision and better practice is clear.
English language appears acceptable. Literature reference is good and so are tables and data.
The title appears precise and in accordance to the article.
Regarding the abstract, it appears to be well structured and contains essential information and salient statistics.
The introduction is brief, clear and touches important points as discussed in precedent and cited articles, which appear to be relevant and of good level and interest.

Experimental design

The utilized method is understandable and appropriate, appearing valid.
The utilized test scores and obtained results also are clear and well discussed.

Validity of the findings

The authors well approached the discussion, underlining the applicability, strengths and also limitations of their study. Further and more detailed studies with great number of patients and a randomized asset should be made in future, based on overcoming limitations.
Future research should focus on randomized controlled trials and long-term follow-up studies to further validate these findings and refine laser treatment protocols for BPH, ensuring broader application and maximized patient benefits.

Additional comments

I therefore believe this article holds a strong point and should represent a starting point for future and stronger studies.
In my opinion no major revision is required.

Reviewer 2 ·

Basic reporting

The grammar and the wording is good enough. There are not many grammar, spelling and punctuation errors.
The references used are relevant and sufficient.
It would be a good idea if the authors could briefly describe how the Greenlight laser functions.
Article structure is acceptable.
The tables and figures are really helpful and necessary for the completion of the authors work.

Experimental design

The authors clearly state in the Introduction section the aim of their study.
The methods are presented in a very extensive way.

Validity of the findings

The authors manage to objectively compare the 180W XPS and the 120W HPS systems and cocnlude on how the 180W surpasses the 120W.
All the necessary data which lead to the conclusions are provided.
From the presented data, the conclusion is complete and represents the work that the authors did.

Additional comments

In the complications sections it would be better if the authors could classify the complications based on the Clavien-Dindo classification

Reviewer 3 ·

Basic reporting

Congratulations for your appropriate use of english.

I suggest you to add some references in the discussion paragraph upon the safety of XPS :
- Marchioni M, Schips L, Greco F, Frattini A, Neri F, Ruggera L, Fasolis G, Varvello F, Destefanis P, De Rienzo G, Ditonno P, Ferrari G, Cindolo L. Perioperative major acute cardiovascular events after 180-W GreenLight laser photoselective vaporization of the prostate. Int Urol Nephrol. 2018 Nov;50(11):1955-1962. doi: 10.1007/s11255-018-1968-9. Epub 2018 Aug 23. PMID: 30141122.
- Ghahhari J, De Nunzio C, Lombardo R, Tubaro A, Brassetti A, De Francesco P, Schips L, Cindolo L. Efficacy and efficiency of Green-Light XPS 180-watt laser system for benign prostatic enlargement in patients treated with 5α-reductase inhibitors. Eur Rev Med Pharmacol Sci. 2021 Jul;25(13):4527-4534. doi: 10.26355/eurrev_202107_26245. PMID: 34286495.
- Castellucci R, Marchioni M, Fasolis G, Varvello F, Ditonno P, Di Rienzo G, Greco F, Altieri VM, Frattini A, Ferrari G, Schips L, Cindolo L. The safety and feasibility of the simultaneous use of 180-W GreenLight laser for prostate vaporization during concomitant surgery. Arch Ital Urol Androl. 2020 Dec 17;92(4). doi: 10.4081/aiua.2020.4.297. PMID: 33348957.

Experimental design

Define major and minor complications

Validity of the findings

I think that results paragraph should be expanded.

Additional comments

Dear authors,
thank you for submitting this interesting systematic review and meta-analysis to PeerJ.

I found your article interesting and almost suitable for publication.
I think you should only implement the results paragraph as it seems to me it should be expanded, while you reported almost all of your findings in tables.

Furthermore I think you should define minor and major complications.

Reviewer 4 ·

Basic reporting

In this study, the authors conducted a comparative evaluation of the GreenLight 120W HPS and 180W XPS vaporization systems for the treatment of benign prostatic hyperplasia (BPH). They demonstrated that the 180W XPS system offers significant advantages in terms of reducing operative time, lasing duration, and postoperative catheterization time, which is valuable.

Experimental design

1. Holmium and thulium lasers are more commonly used in laser enucleation of the prostate, while the green laser is less frequently employed. Given this context, the authors should elaborate on why they believe this study is important and relevant. Further explanation in the manuscript would help clarify its significance.
2. Line 192: Why were porcine kidney tissues used instead of prostate tissue for evaluation?
3. Line 230: I am unclear on how a larger diameter fiber could reduce the endoscopic field of view while simultaneously increasing the area available for irrigation. Could the authors provide more detail to clarify this point?
4. Is it appropriate to focus on the duration of postoperative catheterization as a key comparison point? In clinical practice, catheter duration often depends on the surgeon's experience. I suggest that the time taken for urine color to return to normal might be a more objective and reliable comparison parameter.
5. Did the authors compare prostate volumes between the two groups? If the 180W group had larger prostates, the greater PSA reduction might not necessarily reflect superior tissue ablation efficiency.

Validity of the findings

Many references used are from before 2019. The authors should consider updating the manuscript with more recent studies, as some of the referenced concepts and theories may now be outdated.

---

## Round 0.2 · accepted · Accept

Congratulations on the Acceptance!

Yours,
Yoshi
Prof. Yoshinori Marunaka, M.D., Ph.D.

Reviewer 2 ·

Basic reporting

The grammar and the wording is good enough. There are not many grammar, spelling and punctuation errors.
The references used are relevant and sufficient.

Experimental design

The methods are presented in a very extensive way.

Validity of the findings

All the necessary data which lead to the conclusions are provided.
The authors have upgraded the quality of the manuscript through the revisions.

Reviewer 3 ·

Basic reporting

Authors adopted a clear and unambiguous professional english.

References adopted open the field on the right background upon the panorama of Green-light adoption for BPH treatment

Tables are sufficient and empower the scientific value of the article.

Experimental design

I have no concern on the experimental desing

Validity of the findings

Finding are valid and interesting and open the field on the appropriate Green-light laser adoption for BPH management.

Additional comments

No additional comments are needed. In my opinion article is suitable for pubblication

Reviewer 4 ·

Basic reporting

The updated manuscript has addressed all my previous comments.

Experimental design

No further questions.

Validity of the findings

The updated manuscript is valued.